# Rural-urban disparity in induced abortion in Ghana: A multivariate non-linear decomposition analysis of the Ghana Maternal Health Survey

Isaac Yeboah[1], Martin Wiredu Agyekum[2], Jerry John Ouner[3], Duah Dwomoh[4], Desmond Klu[5], Mary Naana Essiaw[1], Andrew Kweku Conduah[1], Sarah Asaah Owusu-Kwankye[6]*

**1** Institute of Work, Employment and Society, University of Professional Studies, Accra, Ghana, **2** Legon Centre for Education Research and Policy (LECERP), University of Ghana, Legon, Ghana, **3** Department of Family Health Care Nursing, University of California, San Francisco, California, United States of America, **4** Department of Biostatistics, School of Public Health, University of Ghana, Legon, Ghana, **5** Institute of Heath Research (IHR), University of Health and Allied Sciences, Ho, Ghana, **6** Regional Institute for Population Studies, University of Ghana, Legon, Ghana

* sarahkwankye026@gmail.com

## Abstract

Globally, 73.3 million induced abortions were recorded between 2015 and 2019. There are significant disparities in induced abortions across the rural-urban divide that necessitate targeted policies. In this study, we decomposed the rural-urban disparities in induced abortion in Ghana. Data for the study were extracted from the most recent 2017 Ghana Maternal Health Survey. The sample for this study consisted of women who had ever been pregnant, resulting in a weighted sample of 18,140. A multivariate non-linear decomposition model was employed to decompose the rural-urban disparities in induced abortion. The results were presented using coefficients and percentages. The proportion of women who have had induced abortions in their lifetime was 27.1%. Induced abortion was higher in urban areas (34.1%) than in rural areas (19.4%). Approximately 55 percent of the rural-urban disparities in induced abortion were attributable to differences in women's socio-demographic and obstetric characteristics. Hence, if women's socio-demographic and obstetric characteristics were equalled, the rural-urban disparity in induced abortion would be decreased. Region of residence (25.4%), education (16.6%), and parity (9.4%) explained approximately 51 percent of the rural-urban inequality in induced abortion. This study shows significant rural-urban disparities in induced abortion, with the disparities being attributable to the differences in socio-demographic and obstetrics characteristics: region of residence, education, and parity. Policymakers could focus and work on intensifying sexual and reproductive health educational messages, particularly, among women residing in the middle and southern ecological zone of Ghana, and also targeting the educated.

**Data availability statement:** The datasets generated and analyzed for this study are available in the MEASURE DHS database at the repository; https://dhsprogram.com/data/data-set/Ghana_Special_2017.cfm?flag=1.

**Funding:** This research did not receive any specific grant from funding agencies in the public, commercial, or not-for-profit sectors. The funders had no role in study design, data collection and analysis, decision to publish, or preparation of the manuscript.

**Competing interests:** The author(s) declared no potential conflicts of interest with respect to the research, authorship, and/or publication of this article.

**Abbreviations:** AOR: Adjusted Odds Ratio; GDHS: Ghana Demographic and Health Survey; GHS: Ghana Health Service; GMHS: Ghana Maternal Health Survey; GSS: Ghana Statistical Service; ICPD: International Conference on Population and Development; NMIMR: Noguchi Memorial Institute for Medical Research; SDG: Sustainable Development Goal; UN: United Nations; WHO: World Health Organization.

## Introduction

Abortion is the loss of fetus or embryo before gestation. Abortion could be spontaneous or induced. Spontaneous abortion is the unintentional loss of the embryo or fetus before the 20th week of gestation whereas induced abortion is described as intentional by surgical or medical termination of a live fetus before the time of fetus viability [1]. According to the World Health Organization (WHO), induced abortion can be safe or unsafe [2]. Abortion is safe when it is performed by an "appropriately trained health care provider with methods recommended by World Health Organization (WHO)"; it is unsafe when it is carried out either by a person lacking the necessary skills or in an environment that does not conform to minimal medical standards, or both" [3].

Despite the implementation of several policy frameworks including the 1966 International Covenant on Economic, Social, and Cultural Rights, the 1994 International Conference on Population and Development (ICPD), and the Sustainable Development Goals that emphasizes the importance of women's autonomy and their right to access sexual and reproductive healthcare services including abortion services, the incidence of induced abortion remains a significant public health concern, as most of these abortions are categorized as unsafe [1–3].

Available evidence indicates that nearly 73.3 million induced abortions were reported across the globe yearly between 2015 and 2019 [4,5]. Out of this figure, 45 percent of these abortions were categorized as unsafe, with 97 percent of all unsafe abortions occurring in low- and middle-income countries [1]. Regarding adverse outcomes of induced abortion, 62% of all induced abortion-related mortalities are reported in Africa [6]. In Ghana, 57.5 percent of induced abortions in the country were provided by unskilled providers, thereby making the procedure unsafe [7]. Nevertheless, safe induced abortion is considered a human right of every woman [8]. Hence, accessibility to safe induced abortion reduces the incidence of unsafe abortions and its concomitant health outcomes.

There is a preponderance of evidence showing the factors that influence women to have an induced abortion. Among the factors identified in previous studies include non-use of contraceptives [9], educational attainment level [7], gender preferences [10], marital status and region of residence [11], unmet need for contraception [12], wealth status and parity [13]. Beyond these factors, it is undisputed that place of residence plays a critical role in health service utilization, including the decision to have an induced abortion. Studies conducted in China [14] and India [15] have shown that there are significant disparities in induced abortions across the rural-urban divide. For instance, Rahaman et al. [15] study showed women from poorer households were significantly more likely to have an unsafe induced abortion when they lived in rural areas than when they resided in urban areas. An understanding of the rural-urban disparities is imperative to identify the factors that influence induced abortions exclusively in either rural or urban areas or those that are uniform across both places of residence. This would ensure the development of targeted policies and interventions. Nevertheless, no study in Ghana has disaggregated the associated factors of

induced abortions across the dimension of place of residence. This situation suggests a significant knowledge gap in the current discourse of induced abortion in Ghana. The current study, therefore, seeks to examine the rural-urban disparities in induced abortion in Ghana.

## Theoretical framework

This study is underpinned by the social learning theory. The social learning theory proposes that the environment is a major force in human behavior in that human behavior is influenced by observation, modeling, and imitations [16–17]. An individual learns behavior by observing others in a social setting such as family, friends, teachers, neighbors, and church groups. Individuals assimilate and imitate that behavior when they are associated with rewards or positive experiences. Bandura [17] asserts that reinforcement can account for social learning. Reinforcement is when an individual is aware of prior experience consequences. If the consequence of abortion is positive there is the likelihood of its occurrence and vice versa.

## Methods

### Data source and study design

Data for the study were extracted from the most recent Ghana Maternal Health Survey (GMHS), conducted in 2017. We used the women's individual recode files. The GMHS is a nationally representative cross-sectional survey that uses a two-stage cluster sampling method. The study was restricted to women who have ever been pregnant because pregnancy exposes an individual to the likelihood of induced abortion.

### Outcome variable

Induced abortion was the outcome variable in this study. Women were asked, 'How many pregnancies have ended this way (abortion) in your lifetime?' The response categories yielded responses such as none (0), 1, 2, or more. Those with none were coded as "0=no" and those who had 1 or more were coded as "1=yes".

### Primary exposure

The main explanatory variable was place of residence. The responses for this were "rural" and "urban".

### Potential confounders

The potential confounders were selected based on their association with induced abortion in literature [5,12,13,15,18,19–24,25]. The covariates include age, region of residence, educational attainment, marital status, ethnicity, religion, current contraceptive use, age at first sex, knowledge of ovulation, parity, knowledge of abortion legality, and migration status. The categories of each of the variables are shown in Table 1.

### Statistical analyses

Data for the study were analyzed using Stata version 16 (StataCorp, College Station, Texas, USA). First, the distribution of induced abortion across explanatory variables and covariates was examined using the chi-square test. The results were further disaggregated by place of residence. Performing analysis based on place of residence is underpinned by the social learning behaviour theory that posits that the environment is a major force in human behavior in that human behavior is influenced by observation, modeling, and imitations. Hence, the segregation of the analysis by place of residence is to examine the extent of difference in human behaviour (induced abortion) based on the environment. Multivariable binary logistic regression analysis was carried out to explore the predictors of induced abortion by place of residence. Finally, a multivariate non-linear decomposition analysis was employed to decompose rural-urban disparities in induced abortion.

**Table 1. Univariate and bivariate analysis of induced abortion among ever-pregnant women in Ghana, 2017.**

| Characteristics | Frequency | Percentage (%) | Induced abortion | | p-value |
|---|---|---|---|---|---|
| | | | Yes (%) | No (%) | |
| **Age** | | | | | 0.282 |
| 15-24 | 3194 | 17.6 | 74.0 | 26.0 | |
| 25-34 | 6994 | 38.6 | 72.0 | 28.0 | |
| 35-49 | 7951 | 43.8 | 73.2 | 26.8 | |
| **Place of Residence** | | | | | <0.001 |
| Rural | 8622 | 47.5 | 80.6 | 19.4 | |
| Urban | 9517 | 52.5 | 65.9 | 34.1 | |
| **Region of Residence** | | | | | <0.001 |
| Southern | 10483 | 57.8 | 70.1 | 29.9 | |
| Middle | 5228 | 28.8 | 68.1 | 31.9 | |
| Northern | 2429 | 13.4 | 95.5 | 4.5 | |
| **Educational Attainment** | | | | | <0.001 |
| No education | 4347 | 24.0 | 87.8 | 12.2 | |
| Primary education | 3147 | 17.4 | 73.4 | 26.3 | |
| Middle/JHS | 7251 | 40.0 | 66.9 | 33.1 | |
| Secondary | 2314 | 12.8 | 63.3 | 36.7 | |
| Higher | 1081 | 5.8 | 71.1 | 28.9 | |
| **Marital Status** | | | | | |
| Currently married | 8826 | 48.7 | 79.3 | 20.7 | <0.001 |
| Cohabiting | 4977 | 27.4 | 68.9 | 31.1 | |
| Not in union | 4337 | 23.9 | 64.5 | 35.5 | |
| **Religion** | | | | | |
| Catholic | 1671 | 9.2 | 74.7 | 25.3 | <0.001 |
| Other Christians | 12775 | 70.4 | 68.4 | 31.6 | |
| Muslim | 2786 | 15.4 | 87.7 | 12.5 | |
| Traditionalist/Spiritualist | 391 | 2.2 | 93.4 | 6.6 | |
| No Religion | 516 | 2.8 | 82.9 | 17.1 | |
| **Current Contraceptive use** | | | | | <0.001 |
| Yes | 5405 | 29.8 | 69.9 | 30.1 | |
| No | 12735 | 70.2 | 76.7 | 23.3 | |
| **Age at first Sex** | | | | | <0.001 |
| < 18 years | 10143 | 55.9 | 76.9 | 23.1 | |
| 18 years above | 7997 | 44.1 | 67.1 | 32.9 | |
| **Knowledge of Ovulation** | | | | | <0.001 |
| Wrong Knowledge | 10771 | 59.4 | 76.9 | 23.1 | |
| Correct Knowledge | 7369 | 40.6 | 67.0 | 33.0 | |
| **Parity** | | | | | <0.001 |
| No birth | 1118 | 6.2 | 35.5 | 64.5 | |
| 1-3 | 10348 | 57.1 | 73.7 | 26.3 | |
| 4-5 | 4060 | 22.4 | 74.6 | 25.4 | |
| 6 or more | 2613 | 14.4 | 83.2 | 16.8 | |
| **Knowledge of Abortion Legality** | | | | | |
| Yes | 1705 | 9.4 | 68.3 | 31.7 | 0.002 |
| No | 16435 | 90.6 | 73.4 | 26.6 | |
| **Migration Status** | | | | | |
| Migrant | 11358 | 62.6 | 71.4 | 28.6 | <0.001 |
| Non-Migrant | 6782 | 37.4 | 75.5 | 24.5 | |
| **Total** | 18,140 | 100.0 | 72.9 | 27.1 | |

The multivariate non-linear decomposition analysis partitions disparity in induced abortion into components attributable to changing characteristics of women and due to changing reproductive behaviour of the women. This technique also partitions the two components into segments that represent the contribution of each predictor to each of the two components (C and E) in a detailed decomposition. Component C refers to the part of change attributable to changing reproductive behaviour, while component E denotes disparity attributable to changing characteristics. To take care of the complex nature of the GMHS data, we used the "svyset" command during analysis, and weights and clusters were considered. Sample sizes differ between models due to missing data on included variables. All analysis excluded missing responses and was conducted using Stata version 16.1.

### Ethical approval

The 2017 Ghana Maternal Health Survey was conducted under the scientific guidance of the Ghana Statistical Service, Ghana Health Service (GHS), and Noguchi Memorial Institute for Medical Research (NMIMR). ICF International approved the survey and provided technical assistance. Written Informed consent was obtained from all the respondents before the commencement of interviews with each interviewer.

## Results

### Bivariate analysis of factors associated with induced abortion among women in Ghana

The characteristics of the study population and factors associated with induced abortion are presented in Table 1. Most of the participants were aged 35–49 years (43.8%), 52.5% resided in urban areas and more than half (57.8%) resided in the southern region of Ghana. More than a third of the women had Middle/JHS education (40.0%), close to half (48.7%) were currently married, and approximately 8 in 10 women were Christians (79.6%). A little over half experienced sexual debut before 18 years (55.9%), 70.2% were not currently using contraceptives, 59.4% had wrong knowledge of ovulation, 57.1% had a parity of 1–3 children, 9 in 10 women (90.6%) do not have knowledge of abortion legality and 62.6% were migrants. The prevalence of induced abortion in the study was 27.1%. Induced abortion was higher in urban areas (34.1%) than in rural areas (19.4%). Except for age, there were statistically significant differences in induced abortion across all the characteristics of women (Table 1). Regarding those residing in rural areas, except for migration status, there were statistically significant differences in induced abortion across all characteristics (Table 2). Except for age and knowledge of abortion legality, there were statistically significant differences in induced abortion across all characteristics in urban areas (Table 2).

### Rural-urban disparities in factors associated with induced abortion among women in Ghana

The odds of induced abortion among women residing in the southern regions in both urban (AOR = 2.88; 95%CI = 2.37–3.52) and rural areas (AOR = 3.10; 95%CI = 2.51–3.85) were high compared to those residing in northern regions of Ghana (Table 3). Nevertheless, the odds were higher in rural areas. Compared with women with no education, women with any form of formal education were more likely to have had induced abortions in both urban and rural areas. However, the odds were higher among those with primary (AOR = 1.78; 95%CI = 1.45–2.18) or middle/JHS education (AOR = 2.16; 95%CI = 1.80–2.59) in urban areas. On the other hand, the odds of induced abortion were also higher among those with secondary (AOR = 2.39; 95%CI = 1.81–3.15) or higher education (AOR = 2.29; 95%CI = 1.49–3.53) in rural areas. The likelihood of induced abortion decreased with increasing parity in both rural (AOR = 0.07; 95%CI = 0.05–0.10) and urban areas (AOR = 0.11; 95%CI = 0.08–0.15). Overall, approximately 55 percent of the rural-urban disparities in induced abortion were attributable to differences in women's socio-demographic and obstetric characteristics (Table 4). Therefore, if women's socio-demographic and obstetric characteristics were equalled, the rural-urban disparity in induced abortion would be decreased. Among the socio-demographic and obstetric characteristics, region of residence (25.4%), education (16.6%), and parity (9.4%) explained approximately 51 percent of the rural-urban inequality in induced abortion (Table 5).

**Table 2. Bivariate analysis of induced abortion among women in Ghana stratified by place of residence.**

| Characteristics | Rural (n = 8622) | | Urban (n = 9518) | |
|---|---|---|---|---|
| | Induced abortion Yes (%) | P value | Induced abortion Yes (%) | P-value |
| **Age** | | 0.049 | | 0.732 |
| 15-24 | 14.6 | | 28.5 | |
| 25-34 | 13.4 | | 28.8 | |
| 35-49 | 12.4 | | 28.0 | |
| **Region of Residence** | | <0.001 | | <0.001 |
| Southern | 22.9 | | 33.8 | |
| Middle | 22.1 | | 37.6 | |
| Northern | 4.0 | | 8.3 | |
| **Educational Attainment** | | <0.001 | | <0.001 |
| No education | 5.5 | | 12.5 | |
| Primary education | 14.2 | | 28.3 | |
| Middle/JHS | 22.1 | | 35.5 | |
| Secondary | 26.0 | | 34.0 | |
| Higher | 23.4 | | 23.1 | |
| **Marital Status** | | <0.001 | | <0.001 |
| Currently married | 8.0 | | 20.3 | |
| Cohabiting | 21.1 | | 38.1 | |
| Not in union | 21.8 | | 37.9 | |
| **Religion** | | <0.001 | | <0.001 |
| Catholic | 10.2 | | 28.6 | |
| Other Christians | 19.2 | | 35.8 | |
| Muslim | 4.9 | | 10.7 | |
| Traditionalist/Spiritualist | 3.1 | | 11.1 | |
| No Religion | 8.6 | | 22.1 | |
| **Current contraceptive use** | | <0.001 | | <0.001 |
| Yes | 17.5 | | 32.2 | |
| No | 88.6 | | 26.8 | |
| **Age at first Sex** | | <0.001 | | <0.001 |
| < 18 years | 14.7 | | 34.5 | |
| 18 years above | 10.7 | | 22.5 | |
| **Knowledge of Ovulation** | | <0.001 | | <0.001 |
| Wrong Knowledge | 10.6 | | 25.2 | |
| Correct Knowledge | 18.8 | | 33.2 | |
| **Parity** | | <0.001 | | <0.001 |
| No birth | 54.7 | | 66.2 | |
| 1-3 | 13.3 | | 25.7 | |
| 4-5 | 12.4 | | 26.9 | |
| 6 or more | 7.9 | | 20.5 | |
| **Knowledge of Abortion Legality** | | <0.001 | | 0.263 |
| Yes | 18.2 | | 29.8 | |
| No | 12.9 | | 28.2 | |
| **Migration Status** | | 0.684 | | <0.001 |
| Migrant | 13.3 | | 30.4 | |
| Non-Migrant | 13.0 | | 24.9 | |
| **Total** | **19.4** | | **34.1** | |

**Table 3. Multivariable regression analysis of factors associated with induced abortion among women in Ghana by place of residence.**

| | Pooled | Urban | Rural |
|---|---|---|---|
| | aOR [95%CI] | aOR [95%CI] | aOR [95%CI] |
| **Age** | | | |
| 15-24(ref) | | | |
| 25-34 | 1.93[1.70-2.20]*** | 1.97[1.66-2.34]*** | 1.66[1.36-2.03]*** |
| 35-49 | 2.63[2.28-3.03]*** | 2.42[2.01-2.92]*** | 2.33[1.86-2.93]*** |
| **Region** | | | |
| Northern (ref) | | | |
| Southern | 3.27[2.84-3.77]*** | 2.88[2.37-3.52]*** | 3.10[2.51-3.85]*** |
| Middle | 3.59[3.08-4.18]*** | 3.53[2.87-4.35]*** | 2.92[2.31-3.69] *** |
| **Migration Status** | | | |
| Migrant (ref) | | | |
| Non-migrant | 0.79[0.72-0.86]*** | 0.77[0.69-0.86]*** | 0.87[0.76-1.00]* |
| **Religion** | | | |
| Catholic (ref) | | | |
| Other Christian | 1.13[0.99-1.29] | 0.99[0.82-1.19] | 1.21[0.99-1.49] |
| Islam | 0.77[0.65-0.92]** | 0.54[0.43-0.68]*** | 0.93[0.70-1.23] |
| Traditional/Spiritualist | 0.45[0.28-0.71]** | 0.60[0.26-1.39] | 0.49[0.27- 0.86]* |
| No Religion | 0.76[0.56-1.03] | 0.64[0.41-1.01] | 0.92[0.61-1.40] |
| **Parity** | | | |
| No child (ref) | | | |
| 1-3 | 0.13[0.11-0.16]*** | 0.14[0.12-0.18]*** | 0.12[0.09-0.16]*** |
| 4-5 | 0.12[0.10-0.15]*** | 0.13[0.12-0.19] *** | 0.10[0.08-0.14] *** |
| 6 or more | 0.08[0.06-0.10]*** | 0.11[0.08-0.15] *** | 0.07[0.05-0.10] *** |
| **Knowledge of Ovulation** | | | |
| Wrong knowledge (ref) | | | |
| Correct Knowledge | 1.35[1.24-1.47]*** | 1.31[1.18-1.46]*** | 1.44[1.26-1.65]*** |
| **Legal Abortion** | | | |
| Yes (ref) | | | |
| No | 0.83[0.72-0.95]** | 0.88[0.75-1.03] | 0.77[0.60-0.99]* |
| **Contraceptive Use** | | | |
| Yes (ref) | | | |
| No | 0.73[0.67-0.80]*** | 0.77[0.68-0.86]*** | 0.66[0.57-0.76]*** |
| **Marital Status** | | | |
| Currently married (ref) | | | |
| Cohabiting | 1.45[1.31-1.61]*** | 1.53[1.33-1.76]*** | 1.46[1.24-1.72]*** |
| Not in union | 1.54[1.38-1.71]*** | 1.52[1.33-1.74] *** | 1.43[1.19-1.72]*** |
| **Age at First Sex** | | | |
| Below 18 years (ref) | | | |
| 18 years above | 0.49[0.45-0.54]*** | 0.45[0.40-0.50]*** | 0.54[0.47-0.63]*** |
| **Education** | | | |
| No education (ref) | | | |
| Primary | 1.77[1.53-2.04]*** | 1.78[1.45-2.18]*** | 1.52[1.23-1.87]*** |
| Middle/JHS | 2.22[1.95-2.53]*** | 2.16[1.80-2.59] *** | 1.85[1.53-2.24] *** |
| Secondary | 2.59[2.20-3.04]*** | 2.21[1.79-2.73] *** | 2.39[1.81-3.15] *** |
| Higher | 1.84[1.49-2.27]*** | 1.49[1.15-1.93] ** | 2.29[1.49-3.53]*** |

Note: p value<0.05 = *, p value <0.01**, p value <0.001***.

**Table 4. Overall decomposition of the change in induced abortion.**

| Components | Coefficient | P-value | 95%CI | Percent (%) |
|---|---|---|---|---|
| E | 0.08 | 0.000 | 0.08-0.09 | 55.4 |
| C | 0.07 | 0.000 | 0.06-0.08 | 44.6 |
| R | 0.15 | 0.000 | 0.14-0.16 | 100.0 |

## Discussion

The current study sought to examine rural-urban disparity in induced abortion in Ghana using a multivariate non-linear decomposition analysis. We found that the overall prevalence of induced abortion was 27.1%, which contradicts a previous study that reported an induced abortion of 6.0% in India [18]. Additionally, the findings of this current study are inconsistent with other studies that reported an induced abortion rate of 10.1% in China [26], 13.6% in Ethiopia [19], and 16.0% in Nepal [20]. The high prevalence of induced abortion in this study could be because we limited the data to women who have had an abortion in their lifetime. Also, evidence shows that there has been an increase in contraceptive use in Ghana from 22% in 2014 to 25% in 2017 and an increase in the knowledge of the legality of abortion from 4% in 2007 to 11% in 2017 [21]. Probably, women tend to prevent unwanted pregnancies by using contraceptives, and those who get pregnant use legal means to abort a pregnancy by accessing safe abortion methods or they do not report induced abortion.

Our findings point to rural-urban differentials in induced abortion. Induced abortion was higher in urban areas (34.1%) than in rural areas (19.4%). The findings of this study corroborate other studies that reported a higher proportion of induced abortion in urban areas [18,22]. The result is also inconsistent with previous studies that have reported that rural areas have a disproportionately higher prevalence of induced abortion than urban areas [15]. Evidence shows that in rural areas there is a lack of access to safe abortion health services or procedures. Also, the socio-economic conditions make it very difficult for some people to afford safe abortions. In addition, women in rural areas have a lower degree of autonomy and high unmet need which sometimes lead to unwanted pregnancy and consequently, induced abortion [15,23]. The plausible reason why induced abortion is higher in urban areas than in rural areas could be attributed to high modern contraceptive use in rural areas (27%) compared with 23% in urban areas. Also, high unmet needs in urban areas [21], and the fear of the stigma of being pregnant are mostly for unmarried women. Also, the socio-economic conditions such as the high cost of living in the urban areas compared to the rural areas could contribute to the high induced abortion in urban areas.

Consistent with the findings of a related study by Wei et al. [24], rural-urban disparities in induced abortion were attributable to the differences in parity. The regression analyses show that the odds of induced abortion significantly declined with an increase in parity. This implies that women without a child were more likely to have induced abortions than those with a child (ren). As the number of children increases, the odds of inducing abortion decrease. The situation is the same for the pooled data and the rural-urban data. However, the odds were much lower for those in rural areas than for those in urban areas. Evidence indicates that socio-economic factors, especially education, marital status, and fear of stigma, have been identified as some of the reasons why women without a child may induce pregnancy more than those with a child(ren) [27]. Due to stigma or shame, unmarried women may postpone childbirth by inducing a pregnancy. Culturally, postponing childbearing to a more suitable time, especially after marrying, is a norm that regulates the behaviour of Ghana's rural and urban residents [28,29]. Furthermore, unwanted pregnancy and restriction of family size could explain a decrease in the odds of induced abortion when parity increases.

The education of women emerged as an important factor that explains rural-urban differentials in induced abortion. Greater odds were reported for women with education. Educated women were more likely to induce abortion at all levels than those without education. This was similar for those in rural and urban areas. The results agree with other studies

**Table 5. Multivariate decomposition analysis of factors associated with induced abortion disparity between rural and urban residence.**

| | Due to differences in characteristics | | | Due to differences in the coefficient | | |
|---|---|---|---|---|---|---|
| | Coefficient | P value | Percent | Coefficient | P value | Percent |
| **Age** | | | | | | |
| 15-24 (ref) | | | | | | |
| 25-35 | 0.00 | <0.001 | 2.17 | 0.01 | 0.209 | 4.66 |
| 36-49 | 0.00 | <0.001 | 0.95 | 0.00 | 0.797 | 1.24 |
| **Region** | | | | | | |
| Northern (ref) | | | | | | |
| Southern | 0.03 | <0.001 | 16.62 | 0.00 | 0.615 | −1.72 |
| Middle | 0.01 | <0.001 | 8.78 | 0.00 | 0.254 | 2.54 |
| **Migration Status** | | | | | | |
| Migrant (ref) | | | | | | |
| Non migrant | 0.00 | <0.001 | 1.46 | −0.01 | 0.168 | −3.98 |
| **Religion** | | | | | | |
| Catholic (ref) | | | | | | |
| Other Christians | 0.00 | 0.917 | −0.1 | −0.01 | 0.153 | −7.8 |
| Islam | 0.00 | <0.001 | −1 | −0.01 | 0.005 | −9.41 |
| Traditionalist/Spiritualist | 0.00 | 0.224 | 1.97 | 0.00 | 0.687 | 0.78 |
| No Religion | 0.00 | 0.054 | 1.03 | 0.00 | 0.25 | −1.16 |
| **Parity** | | | | | | |
| No child (ref) | | | | | | |
| 1-3 | −0.04 | <0.001 | −26.32 | 0.01 | 0.248 | 7.03 |
| 4-5 | 0.01 | <0.001 | 7.64 | 0.01 | 0.087 | 6.51 |
| 6 or more | 0.04 | <0.001 | 28.04 | 0.01 | 0.026 | 9 |
| **Knowledge of Ovulation** | | | | | | |
| Correct knowledge (ref) | | | | | | |
| Wrong knowledge | 0.00 | <0.001 | 2.0 | 0.00 | 0.289 | −2.18 |
| **Knowledge of abortion legality** | | | | | | |
| Yes (ref) | | | | | | |
| No | 0.00 | 0.122 | 0.75 | 0.01 | 0.361 | 9.63 |
| **Current contraceptive use** | | | | | | |
| Yes (ref) | | | | | | |
| No | 0.00 | <0.001 | −0.07 | 0.01 | 0.096 | 7.92 |
| **Marital Status** | | | | | | |
| Currently married (ref) | | | | | | |
| Cohabiting | 0.00 | <0.001 | −1.04 | 0.00 | 0.674 | 0.79 |
| Not in union | 0.01 | <0.001 | 3.57 | 0.00 | 0.581 | 0.76 |
| **Age at first sex** | | | | | | |
| Below 18 years (ref) | | | | | | |
| 18 years above | −0.01 | <0.001 | −9.28 | −0.01 | 0.055 | −5.4 |
| **Educational Attainment** | | | | | | |
| No education (ref) | | | | | | |
| Primary | 0.00 | <0.001 | −1.1 | 0.00 | 0.302 | 2.03 |
| Middle/JHS | 0.01 | <0.001 | 7.05 | 0.00 | 0.264 | 3.16 |
| Secondary | 0.01 | <0.001 | 7.94 | 0.00 | 0.662 | −0.35 |
| Higher | 0.00 | 0.002 | 2.75 | 0.00 | 0.09 | −0.66 |

[25,30]. Prior studies reported that education exposes people to many things, including higher knowledge of healthcare, and abortion laws, as well as enabling women to have greater autonomy in their decisions [31–33]. The abortion laws in Ghana frown on the termination of pregnancy. However, a pregnancy can be terminated based on certain factors such as rape and health conditions [31]. The probable reason why educated women in this study induce abortion could be attributed to their knowledge and awareness of the abortion laws, which could influence their decision to induce a pregnancy rather than look for a safe abortion process. In this current study, the odds of induced abortion were higher for women with Middle or Second-cycle education in both the rural and urban areas. Consequently, women in school especially in the middle, secondary, and tertiary may recourse to induced abortion to prevent unwanted pregnancies to enable them to further their education. Biggs et al. [32] argued that women perceived the difficulty of schooling and achieving their career goals when raising a child and this sometimes leads to induced abortion.

We also found that the place of region explains the rural-urban differentials in induced abortion. The results show that women in the Southern and Middle regions were more likely to induce a pregnancy than women in the northern region. Contextually, both the middle and southern regions are more developed than the northern regions. Hence, most of the residents may have autonomy and the socio-economic conditions may influence the decisions of women to induce a pregnancy.

## Strengths and limitations of the study

The GDHS data is nationally representative, making it possible for our results to be generalized. However, our data was limited to women who had induced abortions in their lifetime. Hence, the results should be interpreted in line with lifetime induced abortion, rather than only induced abortion within five years. Again, GDHS is a cross-sectional study that cannot be used to establish causality, but rather, an association (s) between variables. In addition, the self-report measure of the outcome variable, induced abortion, has the potential for unintentional bias.

## Conclusion and recommendation

We found a significant disparity in induced abortion in urban and rural areas. The findings from this study suggest that women in urban areas have more induced abortions than women in rural areas. The difference in the rural-urban disparity is attributed to parity, the education level of women, and the region of residence. Women should be encouraged to use safe abortion procedures, such as accessing health care for abortion services. In addition, women in urban areas should be encouraged to use contraceptives to prevent unwanted pregnancies. There should also be education on the abortion process, as well as easy accessibility of abortion services for both women in rural and urban areas.

## Author contributions

**Conceptualization:** Isaac Yeboah, Mary Naana Essiaw.

**Data curation:** Jerry John Ouner.

**Formal analysis:** Isaac Yeboah, Mary Naana Essiaw.

**Methodology:** Martin Wiredu Agyekum.

**Writing – original draft:** Martin Wiredu Agyekum, Andrew Kweku Conduah, Sarah Asaah Owusu-Kwankye.

**Writing – review & editing:** Jerry John Ouner, Duah Dwomoh, Desmond Klu, Andrew Kweku Conduah, Sarah Asaah Owusu-Kwankye.

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
