## [Decision Letter · Decision Letter 0]

12 May 2025

Dear Dr. Owusu-Kwankye,

Thank you for submitting your manuscript to PLOS ONE. After careful consideration, we feel that it has merit but does not fully meet PLOS ONE’s publication criteria as it currently stands. Therefore, we invite you to submit a revised version of the manuscript that addresses the points raised during the review process.

We look forward to receiving your revised manuscript.

Kind regards,

Margubur Rahaman, M.Phil.

Academic Editor

PLOS ONE

Journal Requirements:

Additional Editor Comments :

The reviewers have highlighted several comments and suggestions which will assist to improve your scientific work. Kindly assure all POLS ONE guidelines during the revision submission.

Reviewers' comments:

Reviewer's Responses to Questions

**Comments to the Author**

1. Is the manuscript technically sound, and do the data support the conclusions?

Reviewer #1: Yes

Reviewer #2: No

2. Has the statistical analysis been performed appropriately and rigorously?

Reviewer #1: Yes

Reviewer #2: Yes

3. Have the authors made all data underlying the findings in their manuscript fully available?

Reviewer #1: Yes

Reviewer #2: Yes

4. Is the manuscript presented in an intelligible fashion and written in standard English?

Reviewer #1: Yes

Reviewer #2: No

Reviewer #1: Thank you so much for providing me the opportunity to review the manuscript entitled "Rural-urban disparity in induced abortion in Ghana: a multivariate non-linear decomposition analysis of Ghana Maternal Health Survey". The manuscript meets all the scientific standards necessary to be published. However, the paper has some minor flaws which need to be addressed before it is suitable for publication.

1. In the abstract section, the same phrase has been repeated. Kindly remove the repeated one.

2. In the multivariate decomposition table, there is no hyphen between the lower and upper limits of CI. Kindly rectify the same.

3. There are a number of variables with missing responses. But you have not mentioned whether you have accommodated missing responses or excluded the same. So, address the same in your methodology section in nutshell.

4. You have put forth an insightful implication of Social Learning Theory. But you have not mentioned is there any specific model which you have used for executing the role of the aforementioned theory on rural urban divide in induced abortion. Kindly elaborate in your methodology section.

5. You have mentioned that you have chosen potential confounders based on prior literature although there is no citation for the same. Please make sure proper citation of the previous studies.

6. Please revise the Table 1 as there is no data for the variable called ‘Knowledge of ovulation’. It seems like you have overlooked the same. If that is not the case please make sure to add a note for the same.

7. Please try to stick on same categories of the variables throughout your writing. For instance, somewhere you have used mathematical sign for parity (6+) nevertheless have used text for the same (6 or more).

Reviewer #2: Introduction

The author wrote on important topic. However, he needs to give clue about induced abortion. Introduction is better to be started with main issue. What is abortion? What is induced abortion? what are major classification of induced abortion like safe or unsafe? What are the positive or negative sides of induced abortion? Thus, your introduction needs re-structuring.

Discussion was poorly written. It needs improvement.

The given justification needs more specific and be clear.

Conclusions

This part also needs more specific and to the point. It seems some conclusions are out of the research findings.

**Do you want your identity to be public for this peer review?** For information about this choice, including consent withdrawal, please see our Privacy Policy

Reviewer #1: **Yes: ** Puja Das

Reviewer #2: No

---

## [Author Response · Author response to Decision Letter 1]

26 May 2025

Dear Editor,

We are very pleased to be offered the opportunity to revise this manuscript. We have responded to the various comments by reviewers below.

Reviewer #1: Thank you so much for providing me the opportunity to review the manuscript entitled "Rural-urban disparity in induced abortion in Ghana: a multivariate non-linear decomposition analysis of Ghana Maternal Health Survey". The manuscript meets all the scientific standards necessary to be published. However, the paper has some minor flaws which need to be addressed before it is suitable for publication.

1. In the abstract section, the same phrase has been repeated. Kindly remove the repeated one. Response: The sentence ‘The results were presented using coefficients and percentages’ appeared twice in the abstract section. One has been removed from the abstract section. Thank you.

2. In the multivariate decomposition table, there is no hyphen between the lower and upper limits of CI. Kindly rectify the same. Response: Thank you for the observation. We found that in the multivariable regression table, there was no hyphen for the age category of the Rural aOR, specifically for those aged 25-34 years. This has been corrected. However, we didn’t present 95%CI for the multivariate decomposition table. Thank you.

3. There are a number of variables with missing responses. But you have not mentioned whether you have accommodated missing responses or excluded the same. So, address the same in your methodology section in nutshell. Response: Thank you for this observation. Sample sizes differ between models due to missing data on included variables. All analysis excluded missing responses and was conducted using Stata version 17.1. This has been incorporated in the methodology section (statistical analyses). Thank you.

4. You have put forth an insightful implication of Social Learning Theory. But you have not mentioned is there any specific model which you have used for executing the role of the aforementioned theory on rural urban divide in induced abortion. Kindly elaborate in your methodology section. Response: Performing analysis based on place of residence is underpinned by the social learning behaviour theory that posits that the environment is a major force in human behavior in that human behavior is influenced by observation, modeling, and imitations. Hence, the segregation of the analysis by place of residence is examine the extent of difference in human behaviour (induced abortion). This has been added to the methods section, specifically statistical analyses.

5. You have mentioned that you have chosen potential confounders based on prior literature although there is no citation for the same. Please make sure proper citation of the previous studies. Response: Thank you for this observation. At the ‘potential confounders’ section we have included all the proper citation in previous studies. Then include [5,12,13,15,18,20-25, 30]. Thank you.

6. Please revise the Table 1 as there is no data for the variable called ‘Knowledge of ovulation’. It seems like you have overlooked the same. If that is not the case please make sure to add a note for the same. Response: This was an oversight. We have provided data for the cross tabulation between ‘induced abortion’ and ‘knowledge of ovulation’. See Table 1.

7. Please try to stick on same categories of the variables throughout your writing. For instance, somewhere you have used mathematical sign for parity (6+) nevertheless have used text for the same (6 or more). Response: Thank you for this observation. For the purposes of uniformity and clarity we used the text ‘6 or more’ for parity throughout the manuscript. Thank you.

Reviewer #2: Introduction

The author wrote on important topic. However, he needs to give clue about induced abortion. Introduction is better to be started with main issue. What is abortion? What is induced abortion? what are major classification of induced abortion like safe or unsafe? What are the positive or negative sides of induced abortion? Thus, your introduction needs re-structuring.

Response: There is a complete overhaul and restructure of the introduction of the manuscript. The new structure of the introduction follows the funnel approach suggested by the reviewer. Thank you. Kindly see the introduction section.

Discussion was poorly written. It needs improvement.

The given justification needs more specific and be clear. Response: All authors have taken a second look at the discussion and revised it accordingly.

Conclusions

This part also needs more specific and to the point. It seems some conclusions are out of the research findings. Response: We have re-read the conclusion and made the necessary conclusion.

---

## [Decision Letter · Decision Letter 1]

23 Sep 2025

Rural-urban disparity in induced abortion in Ghana: a multivariate non-linear decomposition analysis of the Ghana Maternal Health Survey

PONE-D-24-59727R1

Dear Dr. Owusu-Kwankye, 

We’re pleased to inform you that your manuscript has been judged scientifically suitable for publication and will be formally accepted for publication once it meets all outstanding technical requirements.

Kind regards,

Margubur Rahaman, Ph.D.

Academic Editor

PLOS ONE

Additional Editor Comments (optional):

Reviewers' comments:

Reviewer's Responses to Questions

**Comments to the Author**

Reviewer #1: (No Response)

2. Is the manuscript technically sound, and do the data support the conclusions?

Reviewer #1: Yes

3. Has the statistical analysis been performed appropriately and rigorously?

Reviewer #1: Yes

4. Have the authors made all data underlying the findings in their manuscript fully available?

Reviewer #1: Yes

5. Is the manuscript presented in an intelligible fashion and written in standard English?

Reviewer #1: Yes

Reviewer #1: (No Response)

**Do you want your identity to be public for this peer review?** For information about this choice, including consent withdrawal, please see our Privacy Policy

Reviewer #1: **Yes: ** Puja Das

---

## [Editor Report · Acceptance letter]

PONE-D-24-59727R1

PLOS ONE

Dear Dr. Owusu-Kwankye,

I'm pleased to inform you that your manuscript has been deemed suitable for publication in PLOS ONE. Congratulations! Your manuscript is now being handed over to our production team.

Kind regards,

on behalf of

Dr. Margubur Rahaman

Academic Editor

PLOS ONE